# OpenReview forum: "SCERL: A Benchmark for intersecting language and safe reinforcement learning"
_NeurIPS.cc/2022/Workshop/LaReL — LaReL 2022_

### Official Review · Reviewer_QyME · 2022-10-06
**welcome addition of benchmark for safe textual RL, but results difficult to interpret**

**Rating:** 6
**Confidence:** 3

**Review:**

This work proposes a benchmark for safety in textual RL. It extends an existing text-based environment with safety challenges. The authors show that two baselines also used for the original benchmark achieve a high variation in performance and it takes longer for the agents to converge than on the original benchmark.

Pros:

- Welcome addition to an existing benchmark; addressing an important problem, namely safety in textual RL
- a flexible benchmark allowing different types of observations (with warning or without) and allowing different types of penalties
- show that the benchmark is more difficult than the original one with two of the same baselines used in the original benchmark proposal

Cons:

- It's hard to interpret the results without reporting the results of the same agents on the original benchmark. If we would have those results to, we would know exactly what your extensions to the benchmark change in terms of mean performance, number of episodes until convergence, and variance. Now we cannot say much about that.
- You say adding a warning improves mean score for "packing lunch box" but that doesn't seem to be the case from table 1.
- It's quite hard to understand section 3.4 and Figure 3

Some suggestions for writing/style:

- The definition of textual RL in the abstract doesn't seem to have much to do with text.
- I think the motivation in the introduction could be worded a bit differently, not all real-life decision making problems are associated with natural language as suggested in the intro in line 18-19, but of course doing safe RL in terms of not taking violating actions is also important in text-based RL.
- Rewrite line 41-45 a bit, 4 times "new" in a row plus some wrong uses of singular/plural
- s/LaREL/LaReL on page 1 bottom
- y-label cut off Figure 2
- Figure 2 could use some more clear captions, like describing what the legends / lines are referring to
- Same for Figure 3
- Not sure if the two baselines you choose are exactly the ones from Murugesan et al., 2021, but if yes please state this clearer in section 3.2, otherwise I think it's uninformative to compare the performance of the baselines on your safety-extended benchmark to those reported in Murugesan et al., 2021

---

### Official Review · Reviewer_qrBq · 2022-10-19

**Rating:** 6
**Confidence:** 3

**Review:**

This paper presents a safety RL environment where the safety constraints and goals are formulated in natural language. It comes with baselines such as A2C and an agent conditioned on a knowledge base.

Strengths
- Timely and well motivated problem. It is clear that natural language is a valuable modality to express goals and safety constraints. Work that enables a language interface into specifying such constraints will make it easy for humans to communicate their desired constraints.
- The paper is easy to follow. The illustrations are instructive.

Weaknesses
- It seems the description of "unsafe" states is done in a very toyish way where a one-hot attribute of the environment (unsafe) is simply described in textual form as "The environment is in an unsafe state". To me it is unclear where
- Despite the tight space constraints, it would be good to include a dedicated related work section.
- Non-text related RL safety work should be cited and discussed, e.g. Ray et al "Benchmarking Safe Exploration in Deep Reinforcement Learning"
- Legend of figure 2 are hard to read.

Overall, I believe the environment will be of interest to LaReL audience.

---

### Decision · Program_Chairs · 2022-10-20

Accept